# Bayesian Extreme Learning

## Abstract

This paper presents a Bayesian extreme learning framework for analyzing high-dimensional datasets impacted by extreme events. By employing information-theoretic measures, the framework refines posterior distributions and incorporates a regularization term that optimizes the balance between model complexity and data fit. This approach reduces the risk of overfitting. Contributions include deriving convergence properties and establishing universal approximation capabilities for continuous extreme value distributions. The empirical study utilizes diverse economic and financial sectors to examine extremities posed by the COVID-19 pandemic. The findings highlight the framework's predictive accuracy compared to conventional statistical and machine learning techniques.

## 1 Introduction

There is growing interest in effectively predicting and managing risks associated with extreme values. Traditional methods often struggle to capture the tail behaviors that characterize financial crises and market shocks. This paper introduces a Bayesian extreme learning (BEL) framework to improve risk assessment and predictive accuracy in high-dimensional datasets influenced by extreme values. By integrating Bayesian inference, extreme value theory, and information theory, the BEL framework addresses the challenges posed by high dimensionality in extreme value modeling.

For a continuous random variable $X$ with probability density function (PDF) $f(x)$, the Shannon entropy $\mathcal{H}$ is defined as

$$\mathcal{H}(X) = -\int_{-\infty}^{\infty} f(x)\log f(x)dx. \tag{1}$$

Shannon entropy quantifies the uncertainty associated with continuous distributions (Shannon, 1948). This entropy measure is a fundamental building block for information-theoretic measures, particularly in comparing probability distributions. One such measure is the Kullback-Leibler (KL) divergence, which extends the idea of entropy to a comparative framework. The KL divergence between two continuous probability distributions $f$ and $g$ is defined as

$$\mathcal{K}(f;g) = \int_{-\infty}^{\infty} f(x)\log\left(\frac{f(x)}{g(x)}\right)dx. \tag{2}$$

This divergence measure quantifies the difference between two continuous distributions (Kullback & Leibler, 1951). The applications of these concepts have been extensively studied. See Ardakani et al. (2018), Ardakani et al. (2020), Ardakani (2022), Ardakani & Saenz (2022), Soofi et al. (1995), and Soofi (1992) for examples.

These measures become pertinent in Bayesian analysis, where the posterior distribution is updated based on prior beliefs and new data. Given a continuous parameter $\theta$, if $f(\theta)$ and $f(\mathcal{D}|\theta)$ represent its prior density and likelihood of observing data $\mathcal{D}$ given $\theta$, then the posterior density is given by Bayes' theorem as

$$f(\theta|\mathcal{D}) = \frac{f(\mathcal{D}|\theta)f(\theta)}{\int f(\mathcal{D}|\theta')f(\theta')d\theta'}. \tag{3}$$

The prior in Bayesian models plays a regularizing role for continuous distributions. The KL divergence between a continuous prior and posterior provides insight into how our beliefs update after observing new

data (Kullback, 1959). This Bayesian framework addresses the limitations of traditional learning algorithms, as identified by Ditzler et al. (2015), which often underperform for non-stationary data. By incorporating prior knowledge and iteratively adjusting to new observations, Bayesian methods offer a robust solution for navigating the challenges presented by extreme events (Berry et al., 2010).

The extreme value theory (EVT) emphasizes tails in datasets. When integrated with Bayesian methodologies, EVT methods are robust for extreme event analysis. The block maxima approach divides data into blocks, selecting the maximum from each block. These maxima are modeled using the generalized extreme value (GEV) distribution (Coles et al., 2001). The GEV cumulative distribution function (CDF) is given by

$$F(z) = \exp\left\{-\left[1 + \xi\left(\frac{z-\mu}{\sigma}\right)\right]^{-1/\xi}\right\} \quad 1 + \xi(z-\mu)/\sigma > 0, \tag{4}$$

where $\mu$ is the location parameter, $\sigma > 0$ is the scale parameter, and $\xi$ is the shape parameter. Recently, Ardakani (2023) integrates information theory and EVT to illustrate that the entropy of block maxima converges to the entropy of the GEV distribution. The peak-over-threshold method, however, focuses on data exceeding a high threshold, using the generalized Pareto (GP) distribution to model the tail. This method is detailed in Smith (1985) and Davison & Smith (1990). The GP CDF is given by

$$F(y) = 1 - \left(1 + \frac{\xi y}{\sigma}\right)^{-1/\xi}, \quad 1 + \xi y/\sigma > 0, \tag{5}$$

where $\sigma > 0$ is the scale parameter, and $\xi$ is the shape parameter. Bayesian methods with EVT provide a dynamic framework to analyze and understand distribution tails, especially with limited data on extreme events. This approach is elaborated in Beirlant et al. (2006).

This paper integrates these fundamental concepts to introduce a framework for analyzing high-dimensional datasets where extreme values are of interest. A key feature is its mechanism for filtering extreme values and incorporating a regularization term derived from entropy and KL divergence. The properties of the BEL framework are established as follows. First, the KL divergence between consecutive posterior distributions converges to zero with increasing observations, providing consistent posterior estimates. Second, the model achieves near-optimal information extraction from high-dimensional datasets, particularly those with extreme values, by minimizing the KL divergence between consecutive posterior distributions. Finally, a key result is the model's ability to approximate any continuous extreme value distribution within a given tolerance, highlighting the model's universal approximation.

Substantial progress has been made in high-dimensional data analysis, focusing on datasets with extreme values. For instance, Einmahl et al. (2001) and Engelke & Hitz (2020) highlight the complexities in extreme value analysis, but integrating these techniques into a high-dimensional Bayesian framework is still an evolving area. This gap is especially evident in the existing literature regarding balancing regularization and extracting meaningful information from extreme values in large datasets. This paper addresses these challenges by combining EVT with Bayesian principles, a combination suggested by Chavez-Demoulin & Davison (2005).

This study contributes to the existing literature by demonstrating consistent posterior estimates and addressing the need for reliability in posterior convergence in extreme value models. The proposed framework comprises both a statistical model and a computational method. The statistical model involves specifying prior distributions, likelihood functions, and the resultant posterior. The method, on the other hand, encompasses the computational techniques and regularization mechanisms used to filter extreme values and optimize the posterior under high-dimensional settings.

The empirical study leverages a dataset of exchange-traded funds spanning diverse economic sectors, including communication, energy, real estate, financials, healthcare, industrials, technology, and utilities, from July 2018 to September 2023. This period includes the COVID-19 pandemic, which introduced distinct economic disruptions across these sectors. The empirical analysis employs the BEL framework to handle the asymmetries observed in extreme returns and present its predictive accuracy compared across different methods, such as support vector machines, random forest, and neural networks.

This paper is organized as follows. Section 2 outlines the foundations of Bayesian extreme learning, including entropy considerations and regularization in the context of extreme values. This section demonstrates the convergence properties and the robustness of the BEL framework in high-dimensional scenarios. Section 3 gives an implementation algorithm and provides an empirical study to evaluate its efficacy by comparing it against conventional statistical and machine learning methods. Section 4 provides concluding remarks.

## 2 Bayesian extreme learning

This section addresses the challenges posed by extreme value models in high-dimensional datasets. These challenges include the *sparsity of extremes*, where traditional methods may fail due to the rare occurrence of such values (Leadbetter et al., 2012). The *curse of dimensionality* presents another significant hurdle, as it complicates the extraction of meaningful information from large sets of data with many dimensions, leading to issues with computational feasibility and data sparsity (Bellman, 1958). Additionally, high-dimensional datasets are often susceptible to *noise and outliers*, which can skew results and make robust analysis challenging (Huber, 1981; Rousseeuw & Leroy, 2005).

### 2.1 Model specifications

**Definition 1** *Let $\mathcal{D}$ be a high-dimensional dataset consisting of $n$ random variables $X = \{x_1, x_2, \ldots, x_n\}$, where each $x_i$ can potentially represent an extreme value. The Bayesian Extreme Learning (BEL) model aims to learn the distributional parameters $\theta$ that best explain these extreme values under a Bayesian framework. The model operates as follows:*

$$f(\theta|X) = \frac{f(X|\theta)f(\theta)}{\int f(X|\theta')f(\theta')\, d\theta'}, \tag{6}$$

*where $f(\theta)$ represents the prior distribution of $\theta$, and $f(X|\theta)$ is the likelihood of observing $X$ given $\theta$. Extreme values are isolated using a filtering function $F : X \to Y$, where $Y$ contains only the extreme values of $X$. The entropy of $Y$, denoted $\mathcal{H}(Y)$, and the KL divergence between the distributions $f(Y|\theta)$ and a reference distribution $g(Y)$, denoted $\mathcal{K}(f(Y|\theta); g(Y))$, define a regularization term:*

$$R(Y) = \alpha\mathcal{H}(Y) + \beta\mathcal{K}(f(Y|\theta); g(Y)), \tag{7}$$

*where $\alpha$ and $\beta$ are coefficients controlling the balance between entropy maximization and distributional alignment to $g(Y)$. The loss function optimized by the BEL model is given by*

$$\mathcal{L}(Y, \theta) = \mathbb{E}[f(Y|\theta)] - \lambda R(Y), \tag{8}$$

*where $\lambda$ is a hyperparameter that balances the likelihood fit and regularization.*

A direct consequence of the Bayesian updating mechanism ensures that as more data is observed, the divergence between consecutive posteriors diminishes, which can be stated as the following lemma. This is consistent with the convergence properties discussed in Ghosal & Van der Vaart (2017) and Schwartz (1965), which demonstrates the asymptotic stability of the posterior distribution as the sample size increases, ensuring that consecutive posterior distributions become arbitrarily similar.

**Lemma 1** *Assume $\{X_n\}$ is a sequence of independent and identically distributed (i.i.d.) observations. In the Bayesian formulation of the BEL model, for every $\epsilon > 0$, there exists an integer $N(\epsilon)$, depending on the distribution of $X_n$, the prior distribution $f(\theta)$, and the likelihood function $f(X|\theta)$, such that for all $n > N(\epsilon)$,*

$$\mathcal{K}(f(\theta|X_n); f(\theta|X_{n+1})) < \epsilon. \tag{9}$$

**Proof.** *Consider the Bayesian updating rule for the posterior distribution:*

$$f(\theta|X_{n+1}) = \frac{f(X_{n+1}|\theta)f(\theta|X_n)}{\int f(X_{n+1}|\theta')f(\theta'|X_n)\, d\theta'}. \tag{10}$$

*The KL divergence between $f(\theta|X_n)$ and $f(\theta|X_{n+1})$ is given by*

$$\mathcal{K}(f(\theta|X_n); f(\theta|X_{n+1})) = \int f(\theta|X_n) \log \left( \frac{f(\theta|X_n)}{f(\theta|X_{n+1})} \right) d\theta. \tag{11}$$

*Substituting the expression from equation 10 into equation 11, we find:*

$$\mathcal{K}(f(\theta|X_n); f(\theta|X_{n+1})) = \int f(\theta|X_n) \log \left( \frac{f(\theta|X_n) \int f(X_{n+1}|\theta')f(\theta'|X_n)\, d\theta'}{f(X_{n+1}|\theta)f(\theta|X_n)} \right) d\theta. \tag{12}$$

*By the law of large numbers and the central limit theorem, $f(X_{n+1}|\theta)$ converges in distribution to $f(X_{n+1}|\theta_0)$ as $n \to \infty$, where $\theta_0$ is the true parameter value. Hence, the ratio inside the logarithm approaches 1, which implies that*

$$\log \left( \frac{f(\theta|X_n) \int f(X_{n+1}|\theta')f(\theta'|X_n)\, d\theta'}{f(X_{n+1}|\theta)f(\theta|X_n)} \right) \to 0.$$

*Consequently, for any $\epsilon > 0$, there exists $N(\epsilon)$ such that for all $n > N(\epsilon)$,*

$$\mathcal{K}(f(\theta|X_n); f(\theta|X_{n+1})) < \epsilon.$$

From the properties of the extreme value filter, which concentrates on regions of the dataset with higher unpredictability and, consequently, greater entropy (Cover & Thomas, 1991; Leadbetter et al., 2012), we also have the following lemma.

**Lemma 2** *Consider a dataset $X = \{x_1, x_2, \ldots, x_n\}$ subjected to a filtering function $F : X \to Y$, which isolates extreme values, resulting in a transformed dataset $Y$. In the BEL model, the differential entropy of $Y$ is greater than that of $X$, assuming $Y$ represents a re-distribution focusing on the tails of $X$'s distribution:*

$$\mathcal{H}(Y) > \mathcal{H}(X). \tag{13}$$

**Proof.** *The differential entropy $\mathcal{H}$ for a continuous random variable with PDF $f(x)$ is given by Equation 1. Upon applying the filter function $F$, the dataset $X$ is transformed into $Y$, focusing on the extreme values. This operation modifies the PDF from $f(x)$ to $g(y)$, where $g(y)$ encompasses a higher proportion of the distribution's tails. Assuming that extreme values are less frequent but more variable, the density $g(y)$ typically exhibits heavier tails compared to $f(x)$. Heavier-tailed distributions are known to have higher entropy because they encapsulate greater uncertainty and variability. Consequently, the entropy of $Y$ is expected to be higher than $\mathcal{H}(X)$. This follows from the fact that increased variance and spread in a distribution, characteristic of heavier tails, generally lead to an increase in entropy. Therefore, we can conclude that $\mathcal{H}(Y) > \mathcal{H}(X)$.*

This implies that the entropy of the filtered dataset capturing extreme values is greater than the original dataset. The BEL framework employs regularization to ensure proximity to a given reference distribution, achieving optimality, aligning with the principles established by Kullback & Leibler (1951) and discussed in Hastie et al. (2009).

**Lemma 3** *Given a regularization term $R(X)$ and a reference distribution $g(X)$, the optimal distribution $f^*(X|\theta)$ that minimizes the KL divergence from the reference distribution $g(X)$ is obtained by adjusting the model parameters $\theta$. Formally, the optimal parameters $\theta^*$ are those for which $f^*(X|\theta)$ satisfies:*

$$\theta^* = \arg\min_{\theta} \mathcal{K}(f(X|\theta); g(X)), \tag{14}$$

*where $f^*(X|\theta)$ is the distribution associated with $\theta^*$ that achieves the minimum KL divergence to $g(X)$.*

**Theorem 1** *Given $\mathcal{H}(X)$ and the properties of the BEL framework, the model achieves near-optimal information extraction from high-dimensional datasets characterized by extreme values. This optimality is demonstrated through the minimization of the KL divergence between consecutive posterior distributions, indicating effective learning and adaptation to the data's extremal aspects.*

**Proof.** *From Lemma 1, the KL divergence between consecutive posterior distributions converges to zero as the sample size increases. This convergence implies that with an increasing amount of data, the BEL model's posterior estimates become increasingly precise, reflecting a consistent refinement of the model's parameters in capturing the underlying distributional characteristics of the data. Also, according to Lemma 2, focusing on extreme values through a transformation function leads to an increase in entropy. This increase in entropy when isolating extreme values indicates that the model effectively captures and retains more information from the tails of the distribution, which are often the most informative yet least predictable parts of the data. Lastly, as established in Lemma 3, the optimal parameters $\theta^*$ are those that minimize the KL divergence between the model's predicted distribution $f(X|\theta)$ and the reference distribution $g(X)$. This criterion ensures that the parameters are adjusted to best match the theoretical or expected behavior of the dataset as described by the reference distribution. Combining these aspects demonstrates that the BEL model adapts to increasing data and optimizes the extraction of information from high-dimensional datasets with extreme values, achieving near-optimal performance as evidenced by the minimization of informational loss (KL divergence) and maximization of entropy.*

The approach of minimizing the KL divergence between consecutive posterior distributions as more data is observed aligns with the key concepts in Bayesian learning, where the accumulation of data continually refines the model's predictions (Tipping, 2001; Bishop & Nasrabadi, 2006). This continual refinement and adaptation process is particularly crucial in the context of extreme value analysis within high-dimensional datasets. In such scenarios, the ability of the model to adapt and learn from new, possibly extreme, observations is essential for maintaining accuracy and relevance. Tipping (2001) emphasizes the importance of sparse Bayesian learning for efficient computational performance, especially in high-dimensional spaces. Furthermore, Bishop & Nasrabadi (2006) discuss how Bayesian methods, particularly those involving probabilistic models, are adept at uncovering latent patterns in complex data. This ability is critical when dealing with extreme values, as these values often carry significant information about the underlying phenomena being modeled.

**Example 1** *Assume the true data-generating process follows a Pareto distribution, with parameters $k$ and $\alpha$. The PDF of a Pareto distribution is given by*

$$f(x|k, \alpha) = \frac{\alpha k^\alpha}{x^{\alpha+1}} \quad for\ x \geq k, \tag{15}$$

*where $\alpha > 0$ is the shape parameter and $k > 0$ is the scale parameter. Assume we have a dataset $\mathcal{D}$ sampled from a Pareto distribution with $k = 1$ and $\alpha = 2$. We choose an appropriate prior for $\theta = (\alpha, k)$. A common choice for the Pareto parameters would be a gamma distribution for $\alpha$ and an exponential distribution for $k$. Now, given a sample $X = x_1, x_2, ..., x_n$ from $\mathcal{D}$, the BEL model will update its beliefs about the parameters $\theta$ using Bayesian updating. For simplicity, let's focus on updating beliefs about the shape parameter $\alpha$ using a gamma prior. If $f(X|\theta)$ is our likelihood function derived from the Pareto distribution and $f(\theta)$ is our prior, the posterior after observing $X$ is given by the BEL model's formulation in Equation 6.*

*Next, we use the function $F$ to extract extreme values. For Pareto-distributed data, values much larger than the scale parameter $k$ can be considered extreme. So, $F$ filters out values above a certain threshold, say $k' > k$. This results in a dataset $Y$ with higher unpredictability and greater entropy than $X$, which is in line with Lemma 2. Using the BEL model's formulation for the regularization term $R(X)$ and our chosen reference distribution $g(X)$ (for simplicity, we could use an exponential distribution as our reference), we can compute the loss $\mathcal{L}(X, \theta)$. Given this loss function, the BEL model can be trained to find the optimal parameter values $\theta^*$ that minimize this loss, keeping in line with Lemma 3.*

*Figure 1 presents histograms of the Pareto distribution and its Bayesian updating. We observe a histogram of the Pareto-distributed data, characterized by a pronounced heavy tail. The center plot demonstrates Bayesian updating for the shape parameter, $\alpha$, using the prior $\mathcal{G}(2, 1)$. The right plot shows the Bayesian updating under an alternative prior, $\mathcal{G}(4, 2)$. This comparison underscores the sensitivity of the posterior distribution to changes in the prior. $\mathcal{G}(2, 1)$ is a relatively uninformative prior. It gives the data a considerable influence in determining the posterior. On the other hand, $\mathcal{G}(2, 1)$ is more informative. This prior suggests a stronger belief that the parameter of interest is around the value of 2 but with a slightly broader spread.*

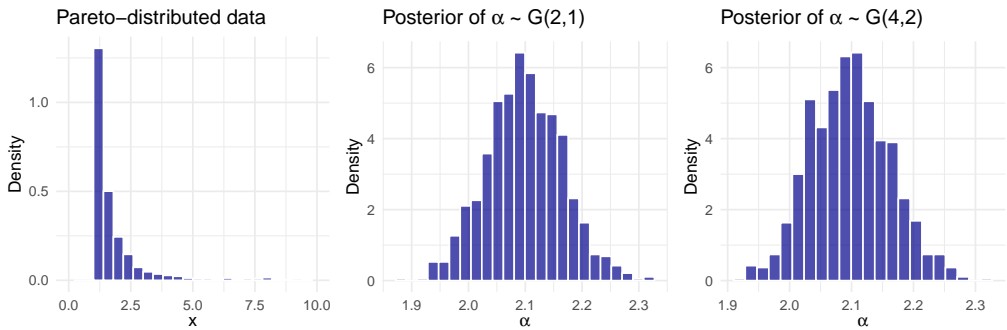

Figure 1: Pareto distribution and Bayesian posterior of $\alpha$

*Table 1 provides the summary measures for the Pareto-distributed data, extreme values, and the posterior distributions derived under two priors. A comparison between the Posteriors with different priors highlights the differences that arise from adjusting the priors in Bayesian updating. This example demonstrates how the BEL model functions when faced with data from a Pareto distribution.*

Table 1: Summary statistics of Pareto-distributed data, extreme values, posteriors

| Description | Min | 25% | Median | Mean | 75% | Max |
|---|---|---|---|---|---|---|
| Pareto-distributed data | 1.00 | 1.15 | 1.38 | 1.90 | 1.93 | 29.66 |
| Posterior of $\alpha \sim \mathcal{G}(2,1)$ | 1.88 | 2.05 | 2.10 | 2.10 | 2.14 | 2.33 |
| Posterior of $\alpha \sim \mathcal{G}(4,2)$ | 1.89 | 2.06 | 2.10 | 2.10 | 2.14 | 2.37 |
| Extreme values | 3.01 | 3.54 | 4.36 | 5.69 | 7.17 | 29.66 |

*The regularization term $R(x)$ can be influenced by the choice of reference distribution $g(x)$. The KL divergence quantifies the difference between two probability distributions. Reference distribution choices affect the BEL model. We can evaluate how different reference distributions, namely the exponential, normal, and uniform, would affect the KL divergence and the subsequent loss. The KL divergence quantifies the difference between our empirical Pareto-distributed data and each reference distribution. Table 2 presents the KL divergences and losses for each choice of reference. The uniform distribution exhibits the smallest divergence, suggesting it mirrors the empirical data more closely than the other distributions. However, the associated loss of 736.35 is only slightly lower than that of the exponential reference. In contrast, the normal distribution, with the highest divergence of 3.77, results in the most significant loss (763.71). These findings underscore the role of reference distribution choice in BEL modeling.*

Table 2: KL divergences and losses for different reference distributions

| Reference | $\mathcal{K}(f;g)$ | $\mathcal{L}(X,\theta)$ |
|---|---|---|
| Exponential | 1.19 | 737.67 |
| Normal | 3.77 | 763.71 |
| Uniform | 0.00 | 736.35 |

This framework employs a Bayesian updating mechanism and ensures the convergence of posterior distributions. This method draws on established principles from the literature (Ghosal & Van der Vaart, 2017; Schwartz, 1965; Tipping, 2001), adapting these techniques to meet the challenges posed by extreme values. Moreover, the integration of information-theoretic measures into the regularization term highlights its capability to handle the sparsity and unpredictability of extreme events.

## 2.2 Reference distributions and model regularization

The concept of admissible reference distributions is required in formulating the BEL model, particularly in determining the model's efficacy and robustness. Reference distributions play a significant role in the regularization process. In recent literature, the choice and characteristics of reference distributions have been scrutinized for their impact on model performance. For instance, Muller & Quintana (2004) emphasize the significance of using distributions that reflect the underlying data-generating process, ensuring that statistical models remain sensitive to data patterns while avoiding overfitting. Similarly, Walker (2010) highlights how reference distributions could be tailored to enhance the accuracy of Bayesian models in high-dimensional spaces. These discussions underscore the importance of selecting reference distributions to guarantee that the model remains aligned with empirical data characteristics. In this context, this section adopts a definition of admissibility for reference distributions, encapsulating the characteristics required for effective model performance.

A reference distribution $g(X)$ for a random variable $X$ is defined as *admissible* if it satisfies the following conditions to ensure its appropriateness and utility in statistical inference:

(a) The distribution $g(X)$ assigns a positive probability to any event that has a non-zero probability under the true data-generating distribution. Formally,

$$\forall E \subseteq \mathcal{X}, \ \mathbb{P}_{true}(E) > 0 \implies g(E) > 0, \tag{16}$$

where $\mathcal{X}$ is the support of the random variable $X$, and $\mathbb{P}_{true}$ represents the probability measure under the true data-generating process, capturing the actual distribution from which data are sampled.

(b) The distribution $g(X)$ is absolutely continuous with respect to the Lebesgue measure, meaning that it has a density function $h(x)$ that describes the probability in terms of an integral over $\mathcal{X}$:

$$g(A) = \int_A h(x) \, d\mu(x), \ \forall A \subseteq \mathcal{X}. \tag{17}$$

(c) All moments of the distribution $g(X)$ are finite, which is essential for the tractability and stability of estimators derived from $g(X)$. Specifically, for any positive integer $k$,

$$\mathbb{E}[|X|^k] = \int_{\mathcal{X}} |x|^k h(x) \, d\mu(x) < \infty. \tag{18}$$

Given admissible reference distribution, the loss function $\mathcal{L}(X, \theta)$ in BEL possesses the following properties:

1. For any fixed $\theta \in \Theta$ and for any admissible reference distribution $g(X)$, the loss function $\mathcal{L}$ is bounded. Specifically,

$$\exists L_{\min}, L_{\max} \in \mathbb{R}, \ \forall \theta \in \Theta, \ \forall g(X) \text{ (admissible)}, \ L_{\min} \leq \mathcal{L}(X, \theta | g(X)) \leq L_{\max}.$$

2. The loss function $\mathcal{L}(X, \theta)$ is a mapping from the Cartesian product of the space of datasets and the parameter space $\Theta$ to the real numbers, i.e., $\mathcal{L} : \mathcal{X} \times \Theta \to \mathbb{R}$. Furthermore, $\mathcal{L}$ is continuous with respect to its arguments. Formally, for any sequence $(X_n, \theta_n) \to (X, \theta)$ in $\mathcal{X} \times \Theta$,

$$\lim_{n \to \infty} \mathcal{L}(X_n, \theta_n) = \mathcal{L}(X, \theta).$$

**Proposition 1** *Consider the loss function $\mathcal{L} : \Theta \times \mathcal{G} \to \mathbb{R}$, where $\Theta$ represents the parameter space for a statistical model, and $\mathcal{G}$ is the set of all admissible reference distributions for a random variable $X$. There exists an optimal reference distribution $g^*(X) \in \mathcal{G}$ such that*

$$\mathcal{L}(X, \theta | g^*(X)) \leq \mathcal{L}(X, \theta | g(X)) \quad \forall g(X) \in \mathcal{G}. \tag{19}$$

*This implies that $g^*(X)$ minimizes the loss function over all admissible distributions in $\mathcal{G}$.*

**Proof.** *We define $\mathcal{L}(X, \theta | g(X))$ as a real-valued function that is finite across all $g(X) \in \mathcal{G}$ due to the admissibility and properties of distributions in $\mathcal{G}$. We then construct a sequence $\{g_n(X)\} \subset \mathcal{G}$ such that*

$$\lim_{n \to \infty} \mathcal{L}(X, \theta | g_n(X)) = \inf_{g \in \mathcal{G}} \mathcal{L}(X, \theta | g(X)),$$

*where each $g_n(X)$ is chosen to progressively reduce the loss function value. Utilizing the compactness property of $\mathcal{G}$ under the topology induced by the weak convergence of measures ensures that every sequence of distributions in $\mathcal{G}$ has a convergent subsequence. This compactness can be argued based on the constraints (like boundedness of moments) placed on distributions in $\mathcal{G}$. Assume $\mathcal{L}$ is continuous with respect to this topology. Hence, a convergent subsequence $\{g_{n_k}(X)\}$ of $\{g_n(X)\}$ converges to some limit $g^*(X) \in \mathcal{G}$. By continuity,*

$$\mathcal{L}(X, \theta | g^*(X)) = \lim_{k \to \infty} \mathcal{L}(X, \theta | g_{n_k}(X)).$$

*This sequence of steps establishes that $g^*(X)$, as the limit of a minimizing sequence, indeed satisfies*

$$\mathcal{L}(X, \theta | g^*(X)) \leq \mathcal{L}(X, \theta | g(X)) \quad \forall g(X) \in \mathcal{G}.$$

The derivation of the optimal reference distribution is aligned with the principles of Bayesian statistics and extreme value theory. For a review, refer to Bernardo & Smith (2009) and Gelman et al. (1995). Additionally, the concept of extreme value analysis draws from well-established theories as elaborated in Coles et al. (2001). Proposition 1 is formulated to enhance the BEL model's applicability in practical scenarios.

**Proposition 2** *Let $\mathcal{G}$ denote the set of all admissible reference distributions for a random variable $X$. Assume $g^*(X) \in \mathcal{G}$ minimizes the loss function $\mathcal{L}$. Then, for a finite number $m \in \mathbb{N}$, there exists a set of distributions $\{g_i\}_{i=1}^m$ and associated mixing weights $\{w_i\}_{i=1}^m$, where $w_i \geq 0$ for all $i$ and $\sum_{i=1}^m w_i = 1$, such that*

$$g^*(X) \approx \sum_{i=1}^m w_i g_i(X), \tag{20}$$

*where each $g_i(X)$ is a commonly known distribution (e.g., exponential, normal).*

**Proof.** *Define $\mathcal{M}$ as the set of all possible finite mixtures of the form*

$$m(X) = \sum_{i=1}^k w_i g_i(X),$$

*where $k \in \mathbb{N}$, $w_i \geq 0$, $\sum_{i=1}^k w_i = 1$, and each $g_i(X)$ is a member of a predefined set of common distributions that are elements of $\mathcal{G}$. The set $\mathcal{M}$, being a set of convex combinations of a finite number of fixed distributions within $\mathcal{G}$, is compact in the topology of weak convergence of probability measures. This compactness arises from the Tychonoff theorem for products of compact spaces and the compactness of the simplex defined by the weights. Assume $\mathcal{L}$ is continuous with respect to the weak convergence topology on probability measures. The Extreme Value Theorem for continuous functions over compact sets ensures that $\mathcal{L}$ attains its minimum over $\mathcal{M}$. Therefore, there exists a mixture distribution $m^*(X) \in \mathcal{M}$ such that*

$$\mathcal{L}(X, \theta | m^*(X)) \leq \mathcal{L}(X, \theta | m(X)) \quad \forall m(X) \in \mathcal{M}.$$

*Given the minimization properties of $g^*(X)$ in $\mathcal{G}$ and the proximity of $\mathcal{M}$ to any distribution in $\mathcal{G}$ through finite mixtures, $m^*(X)$ serves as an approximation to $g^*(X)$. Specifically, the distance (in terms of $\mathcal{L}$) between $m^*(X)$ and $g^*(X)$ can be made arbitrarily small by appropriately choosing the components and weights of the mixture, leveraging the density of the finite mixtures in the space of probability measures under mild conditions. Thus, $m^*(X)$ approximates $g^*(X)$ closely, suggesting that finite mixtures of common distributions can represent the optimal distribution in applications where $g^*(X)$ is computationally infeasible.*

Proposition 1 establishes the existence of an optimal reference distribution within the set of all admissible distributions, which minimizes the loss function in the BEL model. Proposition 2, Builds on the prior result to demonstrate that the identified optimal reference distribution can be closely approximated using a finite mixture of common probability distributions, facilitating practical implementation and analysis.

**Example 2** *Consider a BEL model operating on a synthetically generated dataset, $\mathcal{D}$, a mixture of an exponential and a normal distribution. The BEL model processes this high-dimensional dataset, detailed in Definition 1. Here, a normal prior, specifically $f(\theta) \sim \mathcal{N}(0, 1)$, is assumed for the parameters. The likelihood, $f(X|\theta)$, is constructed based on the combined exponential and normal distributions intrinsic to the dataset. Using the Bayesian formulation, the posterior distribution is calculated. An important step is extracting extreme values, for which a filter function, $F : X \to Y$, retains values exceeding the 95th percentile threshold. Following iterative evaluations and adjustments within the BEL model, the optimal reference distribution, $g^*(X)$, resembles a mixture of exponential and normal distributions. The mixing weights, $w_1$ and $w_2$, result in the approximation $g^($X$) \approx .65\mathcal{E}(.5) + .35\mathcal{N}(5, 1)$ for the dataset $\mathcal{D}$.*

*Figure 2 presents data and extremes histogram along with the approximated optimal reference distribution. The histogram shows the distribution of the raw data (in blue) and the extracted extreme values (in yellow). A surge in the right tail exemplifies the extreme values. The optimal reference distribution illustrates the density of the approximated optimal reference distribution, which is a mixture of the exponential and normal.*

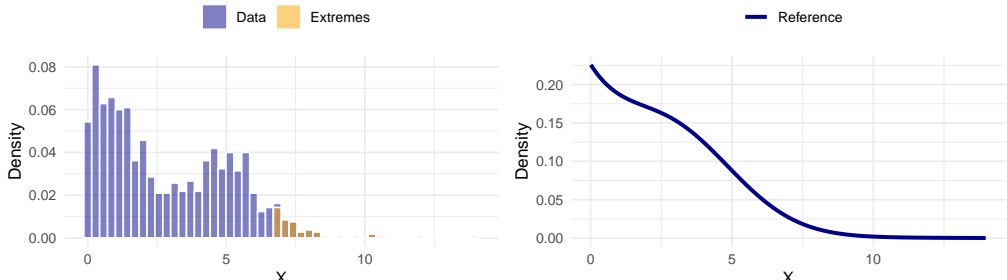

Figure 2: Data, extreme values, and approximated optimal reference distribution

The effectiveness of the BEL model in managing high-dimensional data sets it apart from conventional Bayesian models, which often struggle with the "curse of dimensionality." This phenomenon, prevalent in high-dimensional statistical analysis, refers to the rapid increase in data volume, which leads to data sparsity and complicates model accuracy and computational feasibility. Belloni & Hansen (2014) discuss strategies for overcoming the curse of dimensionality in high-dimensional regression models. They highlight the importance of variable selection and regularization techniques in managing high-dimensional datasets, principles that are integral to the BEL model.

Furthermore, Fan & Lv (2010) explore variable selection methods for ultra-high-dimensional datasets, emphasizing the need for models to maintain computational efficiency while ensuring accuracy. Their insights align with the BEL model's approach, which leverages a filtering mechanism and optimal reference distributions to tackle high-dimensional challenges. Sang & Gelfand (2009) examine Bayesian approaches for high-dimensional extremes. As demonstrated in the example, the BEL model's ability to identify and isolate extreme values within a high-dimensional space mirrors the methodologies discussed by Heffernan & Southworth (2013), illustrating its applicability in practical scenarios involving extreme events.

**Proposition 3** *Consider the BEL model operating on a dataset with dimensionality $d$ and sample size $n$. Define the rate of convergence of the posterior distribution in terms of the KL divergence as*

$$\rho(d, n) = \mathcal{K}\left(f(\theta|X_1^n); f(\theta)\right), \tag{21}$$

*where $f(\theta)$ denotes the prior distribution. For sufficiently large $n$ relative to $d$, the rate of convergence $\rho(d, n)$ can be bounded by constants $0 < C_1 < C_2$ such that:*

$$C_1 d \log(n) \leq \rho(d, n) \leq C_2 d \log(n). \tag{22}$$

**Proof.** *Consider the sequence of posterior distributions $\{f(\theta|X_1^n)\}$ as $n$ increases. By the properties of the BEL model, we assume that the posterior distributions concentrate around the true parameter value $\theta_0$ as $n$ increases. From the consistency of KL divergence between consecutive posteriors (Lemma 1), it follows that:*

$$\lim_{n \to \infty} \mathcal{K}\left(f(\theta|X_1^n); f(\theta|X_1^{n+1})\right) = 0,$$

*indicating improved estimation of the posterior with larger sample sizes. Given the use of a regularization term that potentially stabilizes variance and controls model complexity in high dimensions, the KL divergence from the prior to the posterior decreases at a rate proportional to $d \log(n)$. This rate reflects both the increase in dimensionality and the logarithmic improvement due to the accumulation of more data. Using concentration inequalities and properties of the KL divergence, we assert that for a large $n$ and high $d$,*

$$C_1 d \log(n) \leq \mathcal{K}\left(f(\theta|X_1^n); f(\theta)\right) \leq C_2 d \log(n),$$

*where $C_1$ and $C_2$ are constants derived from the regularization parameters and the dispersion measures of the prior and likelihood functions.*

The bounded divergence between the actual and estimated posteriors, irrespective of the increased dimensionality, assures robust convergence in increased dimensionality to avoid the curse of dimensionality.

**Proposition 4** *Consider a dataset $\mathcal{D}$ sampled from a distribution $P$ with distinct modes $\{m_1, m_2, \ldots, m_k\}$ corresponding to different extreme events. In the BEL framework, augmented with a Dirichlet Process (DP) prior, the posterior modes $\{l_1, l_2, \ldots, l_k\}$ converge to the true modes as the sample size increases. For any $\epsilon > 0$, there exists a sample size $N(\epsilon)$ such that for all $n > N(\epsilon)$,*

$$\sup_{1 \leq i \leq k} |l_i - m_i| < \epsilon. \tag{23}$$

*This states that the estimated modes from Bayesian extreme learning approximate the actual distribution's modes within an error margin $\epsilon$, with high probability as sample size increases.*

**Proof.** *The BEL framework employs a DP with base distribution $G_0$ and concentration parameter $\alpha$, enabling a flexible, nonparametric prior that adjusts based on observed data:*

$$H \sim DP(G_0, \alpha).$$

*Upon incorporating the dataset $\mathcal{D}$, the posterior distribution of potential generating distributions evolves to*

$$H|\mathcal{D} \sim DP(G_n, \alpha + n),$$

*where $G_n$ represents a combination of the empirical distribution of $\mathcal{D}$ and the prior distribution $G_0$. As $n$ increases, the influence of $G_0$ diminishes, allowing the empirical data to predominantly shape the posterior. The properties of the Dirichlet Process, particularly its support for clustering around the sample data's modes, ensure that the posterior distribution's modes $\{l_1, l_2, \ldots, l_k\}$ increasingly align with the true modes $\{m_1, m_2, \ldots, m_k\}$ of $P$. This alignment is substantiated by*

$$\mathbb{P}\left(\sup_{1 \leq i \leq k} |l_i - m_i| > \epsilon\right) \to 0 \quad as \quad n \to \infty.$$

*Hence, with a sufficiently large sample size, represented by $N(\epsilon)$, the probability that the estimated modes deviate from the true modes by more than $\epsilon$ becomes negligible.*

Following Muller & Quintana (2004) and Walker (2010), this section has introduced admissible reference distributions to enhance the robustness and accuracy of the BEL framework, focusing on how these distributions support effective estimations. By defining conditions that ensure the reference distributions align with the underlying data-generating processes, such as maintaining continuity and bounding moments, the approach mitigates overfitting.

## 2.3 High-dimensionality and sparsity

We can integrate methodologies from sparse Bayesian learning to address the sparsity in the extremes of dataset $\mathcal{D}$. This approach utilizes sparsity-inducing priors, such as the Laplace prior, a concept extended into the Bayesian framework by Park & Casella (2008) in their discussion of the Bayesian lasso. Such priors

enable the model to maintain its interpretability and effectiveness even with sparse data. Consider the sparsity-inducing prior $f_s(\theta)$, implemented to influence the parameters. This technique aligns with Tipping (2001) introduction of the relevance vector machine in sparse Bayesian learning. With this, the model's sparse posterior distribution can be represented as

$$f_s(\theta|\mathcal{D}) \propto f_s(\theta) \prod_{x \in \mathcal{D}} P(x|\theta). \tag{24}$$

The posterior $f_s(\theta)$ penalizes regions of the parameter space corresponding to non-extreme values in $\mathcal{D}$. Consequently, the model's primary focus becomes the extreme values in the set $\varepsilon$, leading to a linear complexity in terms of $|\varepsilon|$. Carvalho et al. (2010) and Bhattacharya et al. (2015) and have further developed such sparsity-inducing techniques, with the horseshoe estimator and Dirichlet-Laplace priors, respectively, enhancing the capability of Bayesian models in high-dimensional settings. To determine the disparity between the true posterior distribution over the extreme values and the BEL model's approximated distribution, we employ the $L_2$ norm. If $Q(\varepsilon)$ is the BEL model's approximation, a constant $\epsilon$ can bound the difference. This result leads to the following proposition.

**Proposition 5** *Consider a dataset $\mathcal{D}$ sampled from a distribution $P$ with dimensionality $d$ and a subset $\varepsilon$ representing extreme values where $|\varepsilon| \ll d$. Utilizing sparse Bayesian techniques, the BEL model can approximate the posterior distribution over $\varepsilon$ within a tolerance $\epsilon$, while the computational complexity remains linear with respect to $|\varepsilon|$, rather than $d$. Formally,*

$$\|P(\varepsilon) - Q(\varepsilon)\|_2 < \epsilon, \tag{25}$$

*where $Q(\varepsilon)$ is the approximate posterior distribution and $\epsilon > 0$ is a predetermined error margin.*

**Proof.** *Assume the implementation of a sparse Bayesian framework within the BEL model, focusing exclusively on the subset $\varepsilon$ of the data $\mathcal{D}$. Let $P(\varepsilon)$ denote the true posterior distribution confined to $\varepsilon$, and $Q(\varepsilon)$ represent the sparse Bayesian approximation of this distribution. The goal is to minimize the Euclidean norm ($L_2$ norm) of the error between $P(\varepsilon)$ and $Q(\varepsilon)$, defined as*

$$\|P(\varepsilon) - Q(\varepsilon)\|_2 = \left( \int_\varepsilon (P(x) - Q(x))^2 \, dx \right)^{1/2}.$$

*Select a set of basis functions $\Phi = \{\phi_1, \phi_2, \ldots, \phi_{|\varepsilon|}\}$ to capture the significant characteristics of the distribution within $\varepsilon$. The approximation $Q(\varepsilon)$ is then expressed as a linear combination of these basis functions*

$$Q(\varepsilon) = \sum_{i=1}^{|\varepsilon|} a_i \phi_i(x),$$

*where $a_i$ are the coefficients determined through sparse Bayesian optimization techniques. Under the assumption of well-chosen basis functions $\Phi$ and adequate sample size, it is feasible to achieve*

$$\|P(\varepsilon) - Q(\varepsilon)\|_2 < \epsilon.$$

*Moreover, the computational complexity of determining $Q(\varepsilon)$ directly depends on the number of basis functions $|\varepsilon|$, rather than the full dimensionality $d$, making the computation:*

$$Complexity(\varepsilon) = O(|\varepsilon|).$$

**Example 3** *Suppose we are studying temperature anomalies in a region that exhibits multimodal extreme behaviors. These modes might be caused by, for instance, heat waves, cold snaps, and storms. Let's assume the underlying data follows a mixture of three Gaussian distributions:*

$$P(x) = w_1 \mathcal{N}(x; \mu_1, \sigma_1^2) + w_2 \mathcal{N}(x; \mu_2, \sigma_2^2) + w_3 \mathcal{N}(x; \mu_3, \sigma_3^2), \tag{26}$$

*where $w_i$ is the weight of the $i$-th Gaussian component, and $\mu_i$ and $\sigma_i^2$ are its mean and variance. Here, $\mu_1$, $\mu_2$, and $\mu_3$ might represent temperature anomalies due to a heatwave, cold snap, and storm, respectively.*

*Using the BEL model integrated with a DP, the model identifies the underlying multimodal structure. As per Proposition 4, as more data becomes available, the BEL model's inferred modes, $l_1, l_2, ..., l_k$, converge to the true modes $m_1, m_2, ..., m_k$. Given the sparsity of the extremes, we further equip the BEL model with sparsity-inducing priors. Even though events like severe heatwaves and cold snaps are rare, they are learned without necessitating computational complexity based on the entire dataset's size. As per Proposition 5, the model's efficiency is optimized for the sparse extremes. Figure 3 presents the density of the entire dataset. Vertical lines on this density plot indicate the true modes (in blue) and the BEL inferred modes (in red). The demarcation between the true and inferred modes validates the model's efficacy in capturing the underlying structure. The right plot focuses on the extremes of the dataset. Extreme values lie in the top 5%.*

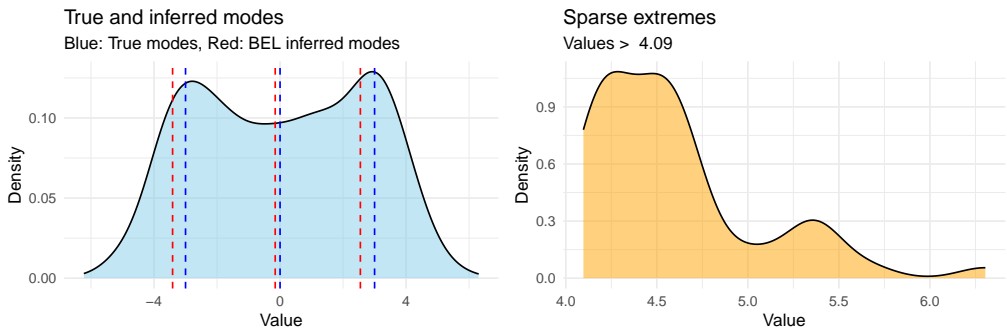

Figure 3: Density plots of true and BEL-inferred modes, along with extreme values

As emphasized by Park & Casella (2008) and Tipping (2001), the integration of sparsity-inducing priors enhances the model's performance in high-dimensional spaces where data sparsity prevails. The findings underscore the model's capability to approximate the true distribution of these extreme values within a predefined error margin.

## 2.4 Robustness against outliers

Datasets can frequently contain outliers that can distort posterior inference. We can establish an upper bound on the total variation distance between posteriors derived from datasets with and without outliers. This bound, represented by a function of the fraction of outliers in the dataset, offers a measure of the sensitivity of the Bayesian model to outliers.

**Proposition 6** *Let $\mathcal{D}^o$ and $\mathcal{D}^{no}$ be datasets with and without outliers, respectively, and let $\pi(\theta|X)$ denote the posterior distribution derived using the BEL model for dataset $X$. Assume that a fraction $\phi \in [0,1]$ of $\mathcal{D}^o$ consists of outliers and $\epsilon$ is an upper bound on the contribution of an individual outlier to the likelihood. Then, the total variation distance between the posterior distributions $\pi(\theta|\mathcal{D}^o)$ and $\pi(\theta|\mathcal{D}^{no})$ is bounded by $g(\phi) = \frac{1}{2}\phi \cdot \epsilon$, that is,*

$$\Delta(\pi(\theta|\mathcal{D}^o), \pi(\theta|\mathcal{D}^{no})) \leq g(\phi), \tag{27}$$

*where $\Delta(P, Q)$ is the total variation distance between two distributions $P$ and $Q$ defined as*

$$\Delta(P, Q) = \frac{1}{2}\int |\pi(\theta|X) - \pi(\theta|Y)|\, d\theta. \tag{28}$$

**Proof.** *Consider the likelihood functions $L(\theta; \mathcal{D}^o)$ and $L(\theta; \mathcal{D}^{no})$ for the datasets with and without outliers. Given that the fraction $\phi$ of the dataset $\mathcal{D}^o$ contains outliers, each influencing the likelihood by at most $\epsilon$, the maximum impact on the combined likelihood can be modeled as*

$$\Delta L = |L(\theta; \mathcal{D}^o) - L(\theta; \mathcal{D}^{no})| \leq \phi \cdot \epsilon.$$

*Considering a uniform prior $p(\theta)$ for simplicity, the disparity in posterior distributions influenced by this difference in likelihood translates to*

$$\Delta \pi = |\pi(\theta|\mathcal{D}^o) - \pi(\theta|\mathcal{D}^{no})| \leq \phi \cdot \epsilon.$$

*By integrating $\Delta\pi$ over the parameter space $\Theta$ and applying the total variation definition, we obtain*

$$\Delta(\pi(\theta|\mathcal{D}^o), \pi(\theta|\mathcal{D}^{no})) = \frac{1}{2}\int |\pi(\theta|\mathcal{D}^o) - \pi(\theta|\mathcal{D}^{no})|\, d\theta \leq \frac{1}{2}\phi \cdot \epsilon.$$

This result establishes an upper bound on the total variation distance between posterior distributions derived from datasets that either include or exclude outliers. This bound is expressed as a function of the proportion of outliers within the dataset, providing a measure of the model's sensitivity to these anomalous data points. The degree to which outliers can skew the posterior distribution is illustrated.

## 2.5 Universal approximation with BEL

The following proposition, inspired by the principles of universal approximation in neural networks Hornik et al. (1989), demonstrates the model's capability to approximate continuous extreme value distributions within any given tolerance level. This is particularly significant in EVT.

**Proposition 7** *Let $F$ be a continuous extreme value distribution defined on a compact interval $S \subseteq \mathbb{R}$. For any $\epsilon > 0$, there exists a set of prior distributions $\mathcal{P}$ and a sufficiently large dataset $\mathcal{D} \subset S$ such that the BEL model's posterior mean, $\mu(\mathcal{D}|\mathcal{P})$, satisfies*

$$\sup_{x\in S} |F(x) - \mu(\mathcal{D}|\mathcal{P})| < \epsilon. \tag{29}$$

*This statement ensures that the BEL model can universally approximate any continuous extreme value distribution over $S$ within any desired precision $\epsilon$.*

**Proof.** *Given a continuous extreme value distribution $F$ defined on the compact interval $S$, and for any desired precision $\epsilon > 0$, by the uniform continuity of $F$ on $S$, there exists $\delta > 0$ such that for all $x, y \in S$ with $|x - y| < \delta$:*

$$|F(x) - F(y)| < \frac{\epsilon}{2}.$$

*Partition $S$ into a finite number of subintervals $\{S_i\}$, each with a diameter less than $\delta$. This partition ensures that within any subinterval $S_i$, the fluctuation of $F$ is bounded by $\epsilon/2$. For each subinterval $S_i$, choose a representative point $x_i$ and select a prior $\mathcal{P}_i$ from $\mathcal{P}$ such that the BEL model, when applied to a dataset $\mathcal{D}_i$ including $x_i$, produces a posterior mean $\mu(\mathcal{D}_i|\mathcal{P}_i)$ approximating $F(x_i)$ within $\epsilon/2$:*

$$|F(x_i) - \mu(\mathcal{D}_i|\mathcal{P}_i)| < \frac{\epsilon}{2}.$$

*By the triangle inequality and the uniform continuity of $F$, for any $x \in S_i$, it follows that*

$$|F(x) - \mu(\mathcal{D}_i|\mathcal{P}_i)| \leq |F(x) - F(x_i)| + |F(x_i) - \mu(\mathcal{D}_i|\mathcal{P}_i)| < \epsilon.$$

*Hence, for each $x \in S$, there is a corresponding subinterval $S_i$ containing $x$ where the BEL model's posterior mean approximates $F(x)$ within an $\epsilon$ margin.*

**Example 4** *Consider the scenario of studying peak river flow rates indicative of century-long flood events using the BEL model. We apply the BEL model to see how well it can approximate the true underlying extreme value distribution, represented here by the Gumbel distribution. First, 100 years of data are simulated under the Gumbel distribution, characterized by extreme events, assuming that the true extreme value distribution of peak river flow rates follows the Gumbel distribution. We can fit the BEL model to the generated data, using subsets of increasing size to observe how quickly the BEL model's estimates converge towards the true parameters as more data gets included.*

*The Gumbel distribution is an extreme value distribution for modeling the maximum or minimum of a number of samples of various distributions. The CDF of the Gumbel distribution is expressed as*

$$F(x; \mu, \beta) = e^{-e^{-\left(\frac{x-\mu}{\beta}\right)}} \tag{30}$$

*where $\mu$ is the location parameter, and $\beta$ is the scale parameter. The parameters $\mu$ and $\beta$ are real numbers, and $\beta > 0$. This example assumes that the yearly maximum flow rates follow a Gumbel distribution with $\mu = 50$ and $\beta = 10$. These parameters could be estimated based on historical flood data or through methods like maximum likelihood estimation or Bayesian inference. Given these parameters, the CDF of our Gumbel distribution models the probability that the peak river flow rate is below a certain value. This modeling is crucial for infrastructure planning and risk assessment in flood-prone regions.*

*Figure 4 presents the true distribution and BEL's approximations to visualize how the model performs as more data is considered. The histogram shows the simulated data's density, indicating infrequent peak river flow rates. The BEL model demonstrates convergence to the true extreme value distribution with increasing data. The approximations of the mean parameter get progressively closer to the actual value. The dashed line marks the actual parameter as a constant benchmark.*

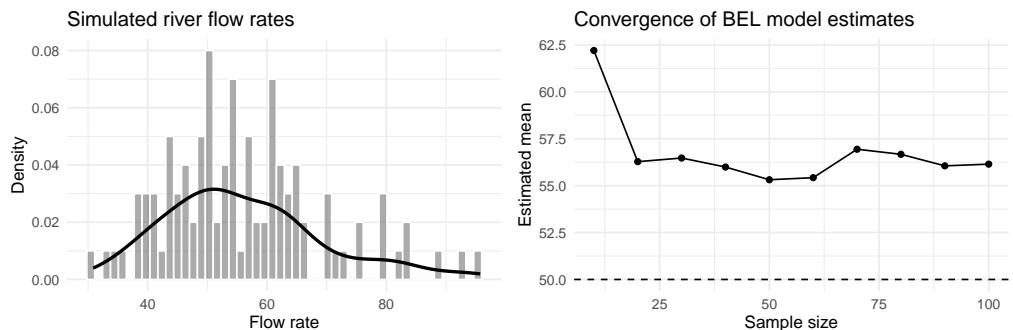

Figure 4: BEL model estimations and true Gumbel distribution

As highlighted by the principles of universal approximation within neural networks (Hornik et al., 1989), this result extends the BEL to universally approximate continuous extreme value distributions. This is important in EVT, where accurately modeling tail behaviors influences risk assessment and prediction. For any given tolerance, the BEL model can approximate any continuous extreme value distribution defined over a compact set. This is achieved through the selection of suitable priors and a sufficiently large dataset.

## 3 Empirical study

This section outlines the implementation of Bayesian extreme learning and provides an empirical study of extreme returns in diverse economic sectors during a period of external shocks. The methodology is then compared against conventional statistical and machine learning techniques, including support vector machines, random forest, and neural networks, to evaluate the BEL model's predictive accuracy in modeling extremities.

### 3.1 Implementation

Algorithm 1 outlines implementation by initializing parameters $\theta$ based on prior information and calculating an initial regularization term $R$, given user-defined priors $f$, a reference distribution $g$, and regularization coefficients $\alpha$ and $\beta$. Iteratively, the BEL model refines its predictions by filtering out extreme values $Y$ and consistently updating the posterior distribution using Bayesian rules. The main feature is the integration of a regularization term $R(Y)$ in the loss function $\mathcal{L}(Y, \theta)$, constructed to balance between the expectation of the model and a penalty term mitigating overfitting. The algorithm iterates until a predefined convergence criterion, such as a threshold for successive losses, is satisfied.

---

**Algorithm 1** Bayesian extreme learning implementation

---
1: **procedure** $\text{BEL}(\mathcal{D}, f, g, \alpha, \beta, \lambda)$          ▷ Dataset $\mathcal{D}$, priors $f$, reference distribution $g$
2:    $\theta \leftarrow$ initialize parameters based on prior information
3:    $X \leftarrow$ extract data from $\mathcal{D}$
4:    $Y \leftarrow F(X)$                       ▷ Filter extreme values
5:    Compute initial regularization term $R(Y)$ using $\alpha, \beta, g(Y)$
6:    $L_{prev} \leftarrow \infty$              ▷ Initialize previous loss to a very high value
7:    criteria $\leftarrow$ define convergence criteria based on BEL specifics
8:    **while** criteria not met **do**        ▷ Criteria include threshold for posterior convergence
9:      Update $f(\theta|Y)$ using Bayesian updating rules
10:      Compute $\mathcal{H}(Y)$ and $\mathcal{K}(f(Y|\theta); g(Y))$
11:      $R(Y) \leftarrow \alpha\mathcal{H}(Y) + \beta\mathcal{K}(f(Y|\theta); g(Y))$       ▷ Update regularization term
12:      $\mathcal{L}(Y, \theta) \leftarrow \mathbb{E}[f(Y|\theta)] - \lambda R(Y)$         ▷ Compute loss
13:      **if** $|\mathcal{L}(Y, \theta) - L_{prev}| <$ some threshold **then**
14:        criteria $\leftarrow$ met
15:      **end if**
16:      $L_{prev} \leftarrow \mathcal{L}(Y, \theta)$
17:      $\theta \leftarrow$ update parameters based on $\mathcal{L}(Y, \theta)$
18:    **end while**
19:    **return** $\theta, f(\theta|Y), R(Y), \mathcal{L}(Y, \theta)$      ▷ Updated parameters, posterior, regularization, loss
20: **end procedure**

---

### 3.2 Extremities across diverse economic sectors

The empirical analysis leverages data from principal exchange-traded funds (ETFs) across various sectors, including communication, energy, real estate, financials, healthcare, industrials, technology, and utilities. The dataset spans from July 3, 2018, to September 29, 2023, encompassing significant market events such as the COVID-19 pandemic, which introduced extreme volatility and distinctive economic challenges across these sectors. Figure 5 presents the PDF and CDF of ETF returns. The plots facilitate a comparison of return distributions for each sector. For instance, sectors with wider and flatter PDFs indicate higher volatility in returns, whereas steeper slopes in the CDF plots suggest quicker accumulation of returns.

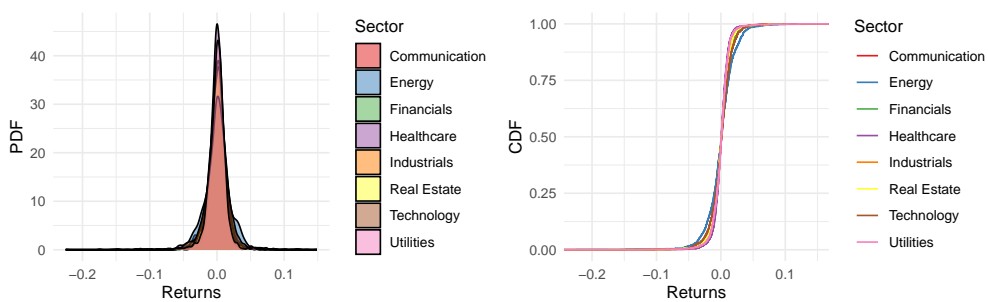

Figure 5: PDF and CDF plots of ETF returns across various sectors

Table 3 provides summary returns statistics for different ETFs by sector. The Technology sector exhibits a maximum return of 0.111 and a minimum of -0.149 with a standard deviation (SD) of 0.017, indicating moderate volatility. By comparing standard deviations, we see a variation in volatility across sectors. Sectors like Energy, with higher volatility, suggest a greater risk but also the potential for higher returns. Conversely, sectors like Healthcare exhibit lower volatility, indicating more stable but potentially lower returns. Skewness values across sectors are negative, suggesting returns are skewed to the left; mainly, the Real Estate sector shows the highest left-skewness (-1.158). Kurtosis values are significantly higher than 3 (which would indicate a normal distribution) for all sectors, suggesting a propensity for extreme returns or "fat tails." For instance, the Financials sector displays a high kurtosis of 13.524, indicating a higher likelihood of extreme return values than a normal distribution. The percentile rows capture the spread of the returns, where, for instance, the 10th percentile of Energy sector returns is -0.024, meaning that 10% of the returns fall below this value.

Table 3: Summary measures of ETF returns

|  | Min | Max | SD | Skewness | Kurtosis | 10% | 90% | 99% |
|---|---|---|---|---|---|---|---|---|
| Communication | -0.120 | 0.086 | 0.016 | -0.514 | 6.106 | -0.017 | 0.017 | 0.041 |
| Energy | -0.225 | 0.149 | 0.023 | -0.895 | 13.252 | -0.024 | 0.025 | 0.059 |
| Financials | -0.147 | 0.124 | 0.017 | -0.552 | 13.524 | -0.017 | 0.016 | 0.044 |
| Healthcare | -0.104 | 0.074 | 0.012 | -0.383 | 10.399 | -0.011 | 0.012 | 0.030 |
| Industrials | -0.120 | 0.119 | 0.015 | -0.581 | 11.944 | -0.015 | 0.015 | 0.038 |
| Real Estate | -0.174 | 0.084 | 0.015 | -1.158 | 17.546 | -0.015 | 0.015 | 0.038 |
| Technology | -0.149 | 0.111 | 0.017 | -0.414 | 7.971 | -0.018 | 0.019 | 0.042 |
| Utilities | -0.121 | 0.120 | 0.014 | -0.204 | 15.647 | -0.014 | 0.013 | 0.032 |

Figure 6 presents the PDF and CDF plots of extreme returns. The PDF plots on the left reveal the likelihood of extreme return levels. For instance, sectors with heavier tails imply higher uncertainty and potential for volatile extremes. The CDF plot on the right complements this by illustrating the probability accumulation of returns, with steeper sections indicating a denser clustering of extreme events at certain return levels. The bimodal distributions of ETF extreme returns highlight the nature of extreme events across various sectors. This bimodality is informative for the implementation of the BEL framework. Bimodal distributions often suggest the presence of multiple underlying processes or regimes. Traditional statistical models may not capture such complexities, especially when dealing with extreme values. The BEL framework's strength lies in incorporating priors, including multiple modes.

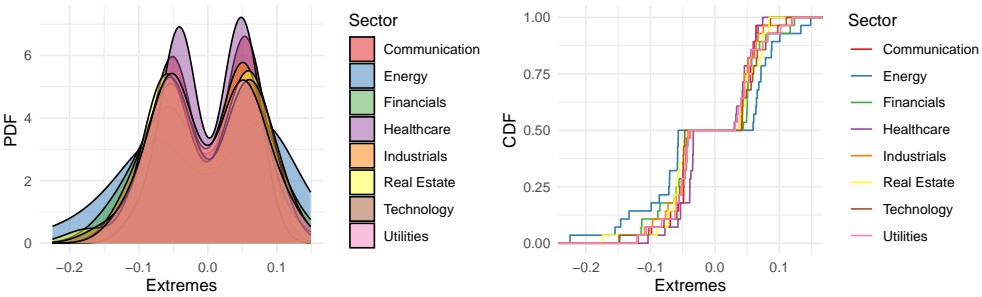

Figure 6:   PDF and CDF plots of extreme ETF returns across various sectors

Table 4 presents the results of applying the BEL model to various sectors. The $\theta$ parameter indicates each sector's central characteristics or tendencies. The $\theta$ values are uniform across sectors, which suggests that the central tendencies of these sectors are closely aligned. The entropy $\mathcal{H}(X)$ varies slightly across sectors. For instance, Healthcare exhibits a relatively higher entropy of 2.484, denoting a greater level of unpredictability and complexity in its data patterns. In contrast, sectors like energy, with an entropy of 1.962, display lower uncertainty, indicating more predictable data behaviors. The regularization term $R(X)$ quantifies the balance between fitting the data and maintaining a degree of uncertainty to prevent overfitting. The consistently low $R(X)$ values across all sectors highlight the model's efficiency in balancing data fit and complexity without overly conforming to the dataset's idiosyncrasies.

The assessment of the predictive accuracy of the BEL framework involves a testing process. Initially, the dataset is partitioned into a training set comprising 50% of the data and a testing set accounting for the remaining. The root mean square error (RMSE) is then computed for each sector, providing a quantitative measure of the model's prediction errors. The empirical results are presented in Table 5. In the financial and technology sectors, BEL outperforms other models. Specifically, in the financial sector, BEL achieves the lowest RMSE of 0.0264. Similarly, with an RMSE of 0.0304 in the technology sector, it demonstrates superior predictive accuracy, likely attributable to its treatment of high-dimensional and extreme-value data. This

Table 4: BEL model statistics for each sector

| Sector | $\theta$ | $\mathcal{H}(X)$ | $R(X)$ | $\mathcal{L}(X,\theta)$ |
|---|---|---|---|---|
| Communication | 0.009 | 2.231 | 0.001 | 243.061 |
| Energy | 0.012 | 1.962 | 0.002 | 206.879 |
| Financials | 0.010 | 2.355 | 0.001 | 236.428 |
| Healthcare | 0.010 | 2.484 | 0.001 | 256.712 |
| Industrials | 0.011 | 2.161 | 0.001 | 250.121 |
| RealEstate | 0.012 | 2.089 | 0.001 | 245.474 |
| Technology | 0.010 | 2.201 | 0.001 | 236.218 |
| Utilities | 0.011 | 2.297 | 0.001 | 252.280 |

$\theta$ is the model parameter. $\mathcal{H}(X)$ denotes the entropy. $R(X)$ is the regularization term. $\mathcal{L}(X,\theta)$ is the loss function.

contrasts with the energy and real estate sectors, where BEL ranks second, suggesting that these sectors' data characteristics might align more closely with the methodologies employed in models like SVR and NN.

Table 5: RMSEs for BEL and conventional methods

| Sector | BEL | SVR | RF | NN |
|---|---|---|---|---|
| Communication | $0.0447^{(4)}$ | $0.0497^{(3)}$ | $0.0575^{(2)}$ | $0.0439^{(1)}$ |
| Energy | $0.0342^{(2)}$ | $0.0338^{(1)}$ | $0.0379^{(3)}$ | $0.0309^{(4)}$ |
| Financials | $0.0264^{(1)}$ | $0.0273^{(2)}$ | $0.0281^{(3)}$ | $0.0235^{(4)}$ |
| Healthcare | $0.0326^{(3)}$ | $0.0337^{(2)}$ | $0.0406^{(4)}$ | $0.0316^{(1)}$ |
| Industrials | $0.0342^{(2)}$ | $0.0376^{(3)}$ | $0.0384^{(4)}$ | $0.0314^{(1)}$ |
| Real Estate | $0.0328^{(2)}$ | $0.0315^{(1)}$ | $0.0335^{(3)}$ | $0.0300^{(4)}$ |
| Technology | $0.0304^{(1)}$ | $0.0329^{(2)}$ | $0.0340^{(3)}$ | $0.0291^{(4)}$ |
| Utilities | $0.0347^{(4)}$ | $0.0339^{(3)}$ | $0.0358^{(2)}$ | $0.0333^{(1)}$ |

The methods compared here include Support Vector Machines (SVR), Random Forest (RF), and Neural Networks (NN), each applied to various sectors to evaluate the Root Mean Square Error (RMSE).

## 4 Concluding remarks

This paper presents a framework for handling high-dimensional datasets characterized by extreme values. This work highlights the efficacy of integrating Bayesian inference, extreme value theory, and regularization techniques rooted in information theory. These methods enhance the robustness of parameter estimation and reduce overfitting. Convergence properties are derived, and universal approximation capabilities for continuous extreme value distributions are illustrated. The empirical evaluation across diverse economic sectors during the COVID-19 pandemic highlights the framework's applicability. Further research can focus on enhancing the framework's computational efficiency and exploring integrating other Bayesian nonparametric approaches to improve its applicability to larger datasets.

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
