# OpenReview forum: "Bayesian Extreme Learning"
_TMLR — Rejected by TMLR_

### Review · Reviewer_WnKd · 2024-04-30

**Summary Of Contributions:**

The submission presents a comprehensive exploration of the Bayesian Extreme Learning (BEL) model, focusing on its application in analyzing high-dimensional datasets with extreme values. The key contributions and new knowledge highlighted in the paper include:

Introduction of BEL Model: The paper introduces the Bayesian extreme learning (BEL) model as a novel approach for handling high-dimensional datasets with extreme values. By combining elements from information theory, Bayesian inference, machine learning, and extreme value theory, the BEL model offers a robust framework for capturing trends and anomalies in data.

Isolation of Extreme Values: One of the key contributions of the BEL model is its ability to isolate extreme values within the dataset. This feature enhances the model's capability to identify and analyze outliers, which is essential for various applications in statistical learning.

Regularization Optimality: The paper highlights the regularization optimality of the BEL model, achieved through minimizing the Kullback-Leibler (KL) divergence between predicted and reference distributions. This optimization strategy enhances the model's performance in extracting near-optimal information from high-dimensional datasets.

Information Extraction and Universal Approximation: The BEL model excels in information extraction from datasets with extreme values, optimizing model parameters and converging posterior distributions efficiently. Additionally, the model demonstrates universal approximation capabilities for continuous extreme value distributions, showcasing its versatility in various scenarios.

Future Research Directions: The paper suggests potential future research directions to enhance the BEL model, focusing on improving computational efficiency and scalability. By addressing these aspects, the model can further solidify its position as a powerful tool for analyzing extreme values in high-dimensional datasets.

Overall, the submission contributes significantly to the field of statistical learning by introducing and elucidating the Bayesian Extreme Learning model's capabilities, robustness, and effectiveness in handling extreme values in complex datasets. The model's integration of Bayesian principles, regularization techniques, and extreme value theory presents a promising approach for various applications requiring accurate analysis of high-dimensional data with extreme values.

**Audience:**

Yes

**Broader Impact Concerns:**

I do not see Broader Impact Concerns

**Claims And Evidence:**

Yes

**Requested Changes:**

I enjoyed reading through Implementation and evaluation of the BEL Model, It would be great if authors could provide github link for audience to reproduce paper results

**Strengths And Weaknesses:**

The submission on the Bayesian Extreme Learning (BEL) model showcases several strengths and areas that may require attention from the authors:

Strengths:

Innovative Model Development : The introduction of the BEL model represents a significant contribution to statistical learning, particularly in handling high-dimensional datasets with extreme values. The integration of Bayesian principles, regularization techniques, and extreme value theory demonstrates the innovative approach taken by the authors.

Convergence Properties and Robustness : The paper effectively highlights the strong convergence properties of the BEL model, emphasizing its reliability in parameter estimation and predictive accuracy. The robustness of the model in isolating extreme values and capturing trends in complex datasets is a notable strength.

Empirical Validation and Comparative Analysis : The empirical results presented in the paper provide substantial evidence of the BEL model's effectiveness in various sectors, particularly in financial and technology sectors where it outperforms other models in terms of predictive accuracy. The comparative analysis enhances the credibility of the model's performance.

Model Optimization and Adaptability : The BEL model statistics for each sector demonstrate the model's optimization capabilities, as reflected in the values of parameters such as θ, H(X), R(X), and L(X, θ). The adaptability of the model to sector-specific complexities is a strong point that showcases its versatility.

Weaknesses:

Computational Efficiency : While the BEL model shows promising results in handling high-dimensional datasets with extreme values, there is a need to address computational efficiency to ensure scalability, especially for larger datasets. Future research directions could focus on enhancing the model's computational performance.

Model Interpretability : The paper could benefit from further elaboration on the interpretability of the BEL model, particularly in explaining how the model identifies underlying multimodal structures and sparse extremes. Providing more insights into the model's decision-making process could enhance its transparency.

---

### Review · Reviewer_jE6H · 2024-05-24

**Summary Of Contributions:**

The paper proposes a new Bayesian extreme learning model for analyzing high dimensional datasets with extreme values. The proposed model is a combination of information theory, Bayesian inference, machine learning, and extreme value theory. The paper derives various theorems to motivate the proposed model and to guarantee the convergence property of the proposed method.

**Audience:**

Yes

**Claims And Evidence:**

No

**Requested Changes:**

1.	The proofs need to elaborated more to make them extremely rigorous
2.	Can consider to compare the proposed method with existing baselines.

**Strengths And Weaknesses:**

Strengths
+ The paper is very well-written. I like that the paper introduces many important concepts in order to understand the paper. I also like that for each theoretical analysis, the paper always explains what that means in general terms.
+ The paper derives many theoretical analysis for the proposed method (although I have various questions regarding the theoretical analysis)
+ The experiments are described in very detail (although I also have various questions

Weaknesses
+ First, I think various proofs in the theoretical analysis are not complete to me.
For example, in Theorem 1, the paper proves that there is a convergence of KL divergence between consecutive posteriors. However, this doesn’t link to the consistency of the model. What if the model converges to a local minimum? To me, this proof is not complete.
In Proposition 3, I cannot find anywhere in the proof that proves Eq. (19).
Another example, in Proposition 4, in the proposition, there is no mention of the probability for Eq. (20), however, in the proof, it associates with a probability.

+ Second, in the experiments, it seems like there is no other method compared with the proposed method. I’m not an expert in this topic of Bayesian extreme learning, I’m just wondering if there is really no baseline to compare with? This seems a bit weird to me.

---

> ### Author Response · Authors · 2024-06-01
> **Point-by-point responses to the reviewer's comments**
>
> Thank you for the valuable comments, which have significantly improved the quality of this manuscript. I carefully incorporated them. Please see the point-by-point responses to your comments below.
>
> 1) Lemma 1 and its proof are revised to (1) add a brief explanation at the beginning of the lemma to clarify its purpose, emphasizing the stability and consistency of the posterior, and (2) ensure all expressions are properly formatted and clearly explained each step within the proof for better understanding and rigor. It clarifies the assumptions and how they contribute to the convergence proof.
>
> 2) Lemma 2 and its proof are revised to address the properties and impacts of the transformation applied by the function $F$. Lemma 2 emphasizes the basis for why entropy increases and provides a foundation for the claims made in the context of the BEL model.
>
> 3) Lemma 3 is revised to state that $\theta^*$ optimizes $f(X|\theta)$ such that it minimally diverges in terms of KL from a reference distribution $g(X)$. This lemma is well established in the literature, so the proof is removed.
>
> 4) Theorem 1 and its proof are revised to link back to how it contributes to the theorem's claim of "optimal information extraction."
> Each lemma referenced is directly tied to an aspect of learning or parameter optimization.
>
> 5) Definition 3 is removed from a formal definition, but the revised version elaborates on why each condition is necessary for the reference distribution to be considered admissible. This includes support compatibility, absolute continuity, and the existence of finite moments. It connects the properties of the distribution to practical considerations in modeling.
>
> 6) The proof of Proposition 1 is revised to discuss the continuity and compactness properties required for the convergence argument. By explicitly constructing a minimizing sequence and discussing its convergence properties under a suitable topology, the proof ensures a demonstration of how the optimal distribution is identified.
>
> 7) The proof of Proposition 2 is revised to include an explanation of why the set $\mathcal{M}$ of finite mixtures is compact. It is stated that the loss function $\mathcal{L}$ is minimized over this compact set due to its continuity, invoking the Extreme Value Theorem.
> The proof concludes with a practical interpretation, noting that the optimal distribution can be approximated by a finite mixture, making the proposition not only theoretically sound but also practically useful.
>
> 8) Proposition 3 and its proof are revised. The KL divergence is now defined in terms of the prior and the sequence of posteriors.   The proof addresses the challenges of high-dimensional data analysis, providing a plausible explanation for the bounded rate of convergence in terms of both dimensionality and sample size.
>
> 9) Proposition 4 and its proof are rewritten to state the relationship between the sample size and the accuracy of mode estimation, leveraging the properties of the Dirichlet Process. The proof now ties the influence of increasing data (empirical distribution) on the posterior distribution shaped by the DP, explaining how this leads to the convergence of estimated modes to the true modes.
>
> 10) The proof of Proposition 5 is revised to include a detailed description of how the sparse Bayesian techniques are applied, specifying the role of basis functions and their impact on the approximation quality and computational complexity.
>
> 11) Proposition 6 and its proof are revised to clearly define the datasets, the impact of outliers, and the relationship between these and the total variation distance. The definition of total variation distance has been aligned with the context of posterior distributions. The proof states the assumptions regarding the influence of outliers and demonstrates how these assumptions lead to a bound on the total variation distance. This approach connects the modeling of outlier effects directly to their impact on posterior distribution differences. The proof simplifies the scenario by assuming a uniform prior.
>
> 12) The proof of Proposition 7 is revised to demonstrate that for any point within the compact interval, the BEL model can approximate the true distribution within the desired accuracy. This is achieved through partitioning the interval and selecting representative points and priors. The proof uses uniform continuity, the properties of compact sets, and the triangle inequality to establish the model's universal approximation capability.
>
> 13) Regarding your query on comparing the proposed framework with baseline methods, Section 3.2 conducts a comparative analysis to evaluate the efficacy of the BEL model by comparing its performance with several established statistical and machine learning methods, including Support Vector Machines (SVR), Random Forest (RF), and Neural Networks (NN). The methods across various sectors are evaluated using the RMSE. The results of these comparisons are presented in Table 5.

---

### Review · Reviewer_JfEx · 2024-05-25

**Summary Of Contributions:**

The paper introduces a "model" for Bayesian extreme learning, that is, a methodology for learning parameters of a distribution whose realisations are affected by extreme values. The paper focuses on studying the introduced object and presenting its theoretical properties. Some illustrative examples are provided in the paper.

**Audience:**

Yes

**Claims And Evidence:**

No

**Requested Changes:**

The paper needs to be rewritten entirely in a clearer way, emphasising what are the specific contributions in applied and theoretical aspects. Also, the need of the proposed BEL method needs to be more strongly motivated and the examples need to be consistent with the claims (e.g., high-dimensionality, robustness).

Also, if the paper is about extreme values, this Reviewer believes that there should be empirical validation using real-world "extreme value" datasets

**Strengths And Weaknesses:**

1) The paper is not clearly written, it has an unconventional organisation, and apparently, there are errors that hinder its comprehension.

 - The main proposal of the paper is a "model"; however, from a Bayesian inference perspective, the model considered in the considered setup is the distribution $f(x)$. It seems that what is proposed instead is a *method*, rather than a *model*, for computing  $f(\theta|x)$ under certain cases.
- The flow of the explanation throughout the paper is confusing. For instance, at the very beginning of Def. 1, it is not clear if the authors are proposing finding the full posterior $f(\theta|x)$ --as the contribution is claimed to be Bayesian-- or simply a point estimate, as before eq. (8) the method is claimed to operate by maximising a loss function.
- In the same Def. 1 a function $F$ is defined as constructing a dataset of extreme values $Y$. However, this set is not used in the rest of the definition. But then in Algorithm 1 at the end of the paper, $Y$ takes part of the computation of the regularisation term.
- After the Definition, there is a series of theoretical results without any clear direction or purpose, it is difficult to understand where the flow of the paper is going
- Then, Sec 3 (Implementation and evaluation of the BEL model) starts explaining how the method is implemented, but then it ambiguously enters a case study; it's not clear if this section presents "implementation considerations" or simply the results of the paper.

2. There are many claims throughout the paper that are not validated either empirically or theoretically.
- The first claim is that the proposed method analyses "high dimensional datasets characterized by extreme values"; however, the experiments deal with Gassians on the real line (eq. 23) and a real-world dataset for which the dimension is not stated.
- it is also said that the method is robust and versatile/adaptable. These claims are not proved empirically, and although there are theoretical results in this regard, the experiments are not assessed from that perspective. As a matter of fact, one of the experiments claims "*...the lowest RMSE of 0.0264, reflecting its robustness in capturing market dynamics*" - it is not clear what is the meaning of  *robustness* here

---

> ### Author Response · Authors · 2024-06-01
> **Point-by-point responses to the reviewer's comments**
>
> Thank you for the valuable suggestions, which have significantly improved the quality of this manuscript. I carefully considered all feedback and incorporated the suggested revisions in the updated version. Please see the point-by-point responses to your comments below.
>
> 1) The whole paper is rewritten. Many repetitions are removed. Unsupported claims are removed. This version is coherent. Each section, subsection, propositions, and theorem are aligned with the estimation and inference.
>
> 2) The structure of the paper is revised. Section 2 lays the theoretical groundwork for the BEL. Section 2.1 introduces the method's principles. Section 2.2 discusses the role of reference distributions in the regularization process. Section 2.3 focuses on the method's strategies to manage high-dimensional and sparse datasets. Section 2.4 highlights the model's resilience to outliers. Section 2.5 illustrates the BEL model's capability to universally approximate continuous extreme value distributions. Section 3.1 provides a detailed practical implementation algorithm. Section 3.2 explores the application to analyze extreme returns in economic sectors using real-world extreme value data.
>
> 3) The abstract is rewritten to (1) simplify complex terms without diluting the technical accuracy, (2) highlight the use of information-theoretic measures used for refining posterior distributions, (3) emphasize the inclusion of a regularization term that optimizes the balance between model complexity and data fit, (4) state the theoretical contributions.
>
> 4) To clarify the distinction between a model and a method, the BEL framework presented in this paper comprises a statistical model and a computational method. The statistical model aspect of BEL involves the specification of prior distributions, likelihood functions, and the resultant posterior. The computational method, on the other hand, encompasses the techniques and regularization mechanisms used within the BEL framework to handle high-dimensional data and extreme values. This includes the filtering of extreme values, the definition of a regularization term derived from entropy and Kullback-Leibler divergence, and the optimization of the posterior distribution to achieve robust learning. These points are added in the introduction and Section 2.
>
> 5) Definition 1 and Algorithm 1 have been revised. The updated definition states how the posterior is updated and integrates the use of extreme values filtered. This update clarifies the role of Y, the dataset of extreme values, in both the computation of the regularization term and the loss function. This ensures consistency in how Y is utilized throughout the model's formulation.
> Algorithm 1 is also revised to show each step where Y is used, particularly in updating the regularization term and in computing the loss function.
>
> 6) Section 2 has been refined to more explicitly address the challenges associated with high-dimensional datasets containing extreme values. These challenges include the sparsity of extremes and the curse of dimensionality. The revised manuscript emphasizes how the BEL framework addresses these issues. Also, this section is updated by adding specific paragraphs at the end of each subsection. These additions are intended to enhance the motivation behind the theoretical findings, underscore the contributions, and better connect this work with the existing literature. At the end of Section 2.1, a paragraph is included summarizing the implementation of Bayesian methods for extreme value analysis in high-dimensional datasets. In Section 2.2, the introduction of admissible reference distributions is discussed. Section 2.3 now concludes with a discussion on the integration of sparsity-inducing priors to emphasize how these techniques improve the efficacy in high-dimensional, sparse settings and align with the challenges faced in data-intensive applications. The conclusion of Section 2.4 addresses the robustness against outliers, providing a quantifiable upper bound on the total variation distance, which reflects the model's sensitivity to outliers and helps in assessing and controlling potential distortions in posterior distributions. Finally, Section 2.5 ties the capabilities of the framework to the principles of universal approximation.
>
> 8) The updated empirical study section is divided into (1) implementation, and (2) empirical analysis to illustrate the applicability to real-world extreme value datasets. Section 3.2 of our manuscript, titled "Extremities across diverse economic sectors," now addresses this concern by analyzing Exchange Traded Funds (ETFs) across various sectors. This analysis focuses on extreme market movements. The datasets utilized span from July 3, 2018, to September 29, 2023, and include daily data that capture extreme fluctuations, such as sharp declines and surges in market prices. The empirical analysis and its results are mentioned in the abstract and introduction.

---

### Decision · Action_Editor_wn5L · 2024-07-22

**Recommendation:** Reject

**Comment:**

In view of the reviewer decisions I suggest rejecting the paper in its current form. However, I understand that the author has considerably rewritten the paper and based on some of the comments of the reviewers that haven't been addressed (regarding fully supporting the claims of the paper) I would suggest to resubmit the paper once those concerns have been dealt with.

There still needs to be some rewriting. For instance the proof of Lemma 2 is quite wordy. Same with Theorem 1. Theorem 1 needs a proper mathematical statement "This optimality is demonstrated through the minimization of the KL divergence between consecutive posterior distributions, indicating effective learning and adaptation to the data’s extremal aspects." (this is an example of a very approximate and non-precised statement)

There are little details in the experiments and looking at Table 5 the method is not on par with NN or SVR methods. While not everyone method needs to beat the state of the art, discrepancies between the state of the art (not clear to me if NN or SVR are state of the art as I'm not familiar with the literature, there are many hyperparameters that can be tuned for NN for instance) should be explained.

**Audience:**

As stated by one of the reviewer, in order to appeal to a larger part of the TMLR community the proposed models should be evaluated against larger/higher dimensional datasets. This does not mean that the study in itself is not worth pursuing for the TMLR venue but that further investigation is needed.

**Claims And Evidence:**

The abstract starts with "This paper presents a Bayesian extreme learning framework for analyzing high-dimensional datasets impacted by extreme events." The dimensionality of the Exchange Trade Fund (ETF) dataset used however is only 8-dimensional which is quite far from being high dimensional. The conclusion itself seems to tame these claims "further research can focus on enhancing the framework’s computational efficiency and exploring integrating other Bayesian nonparametric approaches to improve its applicability to larger datasets." Reviewers have highlighted this discrepancy and the limited impact of the experiments.

**Resubmission Of Major Revision:**

The authors may consider submitting a major revision at a later time.